# Changes Caused by Bisphenols in the Chemical Coding of Neurons of the Enteric Nervous System of Mouse Stomach

**DOI:** 10.3390/ijerph20065125

**Published:** 2023-03-14

**Authors:** Krystyna Makowska, Slawomir Gonkowski

**Affiliations:** 1Department of Clinical Diagnostics, Faculty of Veterinary Medicine, University of Warmia and Mazury in Olsztyn, Oczapowskiego 14, 10-957 Olsztyn, Poland; 2Department of Clinical Physiology, Faculty of Veterinary Medicine, University of Warmia and Mazury in Olsztyn, Oczapowskiego 13, 10-957 Olsztyn, Poland

**Keywords:** bisphenol A, bisphenol S, mice, enteric nervous system, stomach, neuroplasticity, immunofluorescence labeling

## Abstract

Bisphenol A (BPA), an organic chemical compound which is widely used in the production of plastics, can severely damage live organisms. Due to these findings, the plastic industry has started to replace it with other substances, most often with bisphenol S (BPS). Therefore, during the present investigation, with the use of double immunofluorescence labeling, we compared the effect of BPA and BPS on the enteric nervous system (ENS) in the mouse corpus of the stomach. The obtained results show that both studied toxins impact the amount of nerve cells immunoreactive to substance P (SP), galanin (GAL), vesicular acetylcholine transporter (VAChT is used here as a marker of cholinergic neurons) and vasoactive intestinal polypeptide (VIP). Changes observed under the impact of both bisphenols depended on the neuronal factor, the type of the enteric ganglion and the doses of bisphenols studied. Generally, the increase in the percentage of neurons immunoreactive to SP, GAL and/or VIP, and the decrease in the percentage of VAChT-positive neurons, was noted. Severity of changes was more visible after BPA administration. However, the study has shown that long time exposure to BPS also significantly affects the ENS.

## 1. Introduction

Bisphenol A (BPA) is an organic chemical compound of the phenol family that is widely used in the production of plastics [1]. However, more and more studies have shown the many negative effects of BPA on the living organism. Until now, the harmful effects of this substance have been noticed in almost every internal system of the organism, first of all including the endocrine, reproductive and nervous systems [2].

These allegations have led to different ways of reducing the use of BPA in the plastic industry. Therefore, many countries have introduced various restrictions on the use of BPA [1,3] and several substitutes of BPA have been implemented for the plastic production. The substitute that is most commonly used is bisphenol S (BPS) [4]. Although at first BPS seemed safe for living organisms, more and more studies have featured its harmful effects on humans and animals [4,5,6]. It has been discovered that, as with BPA, BPS is also an endocrine disruptor with a negative impact on endocrine, digestive and reproductive systems [7].

Bisphenols can get to the organism through several routes, namely through the digestive tract, airway or skin [8]. The main route of exposure is the gastrointestinal tract, which is connected with the penetration of bisphenols from food containers and bottles to food and drinking water [1]. In the living organism, every alimentary toxin has to cross the intestinal barrier formed with the immune system and enteric nervous system (ENS) [9]. Therefore, the first subclinical signs of bisphenol intoxication can involve disturbances in the digestive tract.

The knowledge of the bisphenol impact on the digestive system is relatively scant [1,2,10,11,12,13,14,15], and the least is known about the effects of these substances on the ENS located in the gastric wall. Namely, in the light of previous studies, it is only known that BPA may affect the enteric neurons in the stomach of the domestic pig [16], and BPA and BPS change the number of nitrergic neurons in the gastric wall in mice [17], but the influence of these substances on other population of neurons in the gastric ENS in the rodents is unknown.

The ENS in the mouse stomach forms two intramural ganglia. The first of them, placed in the submucosal layer, is the submucosal ganglia (SG), and the second is the myenteric ganglia (MG), located in the muscle layer [18].

Because of the millions of neurons located in the enteric ganglia and the considerable independence from the central nervous system, the ENS is commonly called the “intestinal” or “second” brain [19]. The ENS regulates the gastrointestinal tract functions, not only in physiological conditions, but also taking part in the response to pathological stimuli. Enteric neurons are able to undergo morphological and functional variations, including changes in their neurochemical coding [20,21,22]. This term includes the ability of nerve cells to produce and release neurotransmitters. So far, dozens of such biologically active substances have been described in the ENS, and some of the most important neuronal factors produced by enteric neurons are acetylcholine, galanin (GAL), vasoactive intestinal polypeptide (VIP) and substance P (SP) [23].

It should be pointed out that the ENS, together with gastrointestinal immune cells, is strictly functionally connected with the gastrointestinal barrier, constituting the first line of defense against pathological and toxic agents entering the body through food [9]. It is also known that changes in the ENS may be the first signs of the action of such factors, which appear before other symptoms [24,25]. One of the first and most important signs of the action of pathological factors on the digestive system is a change in the neurochemical characterization of the enteric neurons [20].

Therefore, the aim of present investigation was to evaluate and compare, for the first time, changes in the percentage of the enteric neurons in the mouse stomach containing selected neuronal factors, such as GAL, VIP, VIP and vesicular acetylcholine transporter (VAChT is used here as a marker of cholinergic neurons) caused by the administration of various doses of BPA and BPS. The selection of the stomach is not accidental. It is the first segment of the digestive tract, where the food content stays for a relatively long time, and therefore the stomach is significantly vulnerable to toxic factors in the food. Moreover, it is known that BPA negatively affects the stomach, causing the decrease in the gastric mucosal thickness, and may contribute to the formation of gastric ulcers [26], but the mechanisms of this phenomenon are unknown.

The results obtained during this study will contribute to the better understanding of the influence of bisphenols on the innervation of the mammalian stomach. Moreover, the present research will help to determine whether BPS acts in a similar way to BPA on the ENS in the stomach.

## 2. Materials and Methods

A total of 35 CD1 strain mice, both males and females, were used in the present investigation. Throughout the whole experiment, animals were kept under standard laboratory conditions, including constant temperature at 22 ± 2 °C, humidity at 55 ± 10%, 12:12 h light–dark cycle, and water and food *ad libitum*.

All experimental activities were approved by the Local Ethical Committee for Animal in Experiments in Olsztyn at the University of Warmia and Mazury in Olsztyn (Poland), and the number of agreement was 46/2019. Moreover, the experiment was performed according to Act for the Protection of Animals for Scientific or Educational Purposes of 15 January 2015 (Official Gazette 2015, no. 266), applicable in the Republic of Poland. During this experiment, all methods were carried out in accordance with relevant guidelines and European and Polish regulations. Moreover, the study was carried out in compliance with the ARRIVE (Animal Research: Reporting of In Vivo Experiments) guidelines.

At the age of 3 months, when the animals gained about 30 g body weight (b.w.), mice were randomly divided into 5 groups (7 mice per group). The arrangement of the animal groups in the experiment was as follows: (1) control group (C group, *n* = 7)—animals were not subjected to any treatment; (2) BPAI group (*n* = 7)—animals received BPA at a dose of 5 mg/kg b.w./day; (3) BPAII group (*n* = 7)—mice were exposed to BPA at a dose of 50 mg/kg b.w./day; (4) BPSI group (*n* = 7)—animals were treated with BPS at a dose of 5 mg/kg b.w./day; (5) BPSII group (*n* = 7)—mice received BPS at a dose of 50 mg/kg b.w./day.

Bisphenols were administered in the same manner in all animal groups for 3 months. Bisphenols were dissolved in the drinking water, and the method of administration of the compounds was established according to the method previously described by Dobrzynska et al. 2018 [27] and Rezg et al. 2018 [28]. Due to the fact that bisphenols are insoluble in water, compounds were dissolved in 20 µL of ethyl alcohol (70%) and then added to drinking water. Only alcohol was added to the water for the control animals in the same volume as for the other groups.

The justification for the doses of bisphenols used is that, in the light of previous studies, the lower dose of BPA used in the present study (5 mg/kg b.w.) is regarded as the no-observed-adverse-effect-level (NOAEL) dose, and the higher dose (50 mg/kg b.w.) as the lowest-observed-adverse-effect-level (LOAEL) dose in mice [29,30,31]. Both bisphenols were administered in the same doses to compare their influence on the ENS. Every week, animals were weighed to establish the appropriate doses of bisphenols.

After 3 months, all mice were decapitated and fragments of the corpuses of stomachs (glandular stomachs) were collected immediately after the death of the animals. Then, organs were put in the 4% buffered paraformaldehyde (pH 7.4) for 24 h. For the next three days, the stomachs were rinsed using a phosphate buffer (0.1 M, pH 7.4, at 4 °C), then, for at least 3 weeks, the tissues were kept in an 18% phosphate-buffered sucrose solution at 4 °C. Next, the tissues were frozen at −22 °C, cut with the use of a cryostat (Microm, HM 525, Walldorf, Germany) to 12 μm thick sections and fixed on microscope slides.

Stomach fragments were subjected to standard double immunofluorescence labeling, as described in the previous literature [18]. This technique consisted of several steps (all steps performed at room temperature—rt): (1) Drying at room temperature (rt) for 1 h; (2) Blocking (1 h) using a horse serum solution (10% horse serum, 0.1% bovine serum albumin, 0.1 MPBS, 1% Triton X-100, 0.05% thimerosal, 0.01% sodium aside); (3) Incubation in a humid chamber with a mixture of primary antisera (overnight). Specification of antisera used in the present study is shown in the Table 1; (4) Incubation (1 h) with a mixture of species-specific secondary antibodies conjugated with Alexa fluor (Table 1) in order to visualize the complexes of antigen–primary antibody. Between the above-mentioned steps, the tissues were rinsed in PBS (3 × 10 min).

To eliminate the nonspecific staining, routine specificity tests of the antibodies used in the study were conducted. Those tests included the preabsorption of antibodies with the appropriate antigen, an omission test and the replacement of primary antibodies by nonimmune sera. Moreover, the specificity of antibodies included in this study on the mice tissues was checked in previous studies (Table 1).

The analysis of labeled tissues was performed using the immunofluorescence microscope Olympus BX51 (Olympus, Tokyo, Japan) with appropriate filter settings combined with an Olympus XM10 camera (Olympus, Tokyo, Japan).

During the present study, the evaluation of the percentage of neurons containing VIP, GAL and/or SP was performed. To this end, at least 500 neuronal cells immunoreactive to the pan-neuronal marker protein gene product 9.5 (PGP 9.5) from each animal located in each type of the enteric ganglia were examined for the presence of each of the neuronal factors mentioned above. Only cells with a good visible nucleus were included in the study. The number of PGP-9.5-positive cells was regarded as 100%. In order not to count the same perikaryon twice, the studied tissue fragments were located at least 100 µm apart. The obtained results were pooled and presented as mean ± SEM.

To determine whether bisphenols affect the total number of the enteric neurons in the gastric wall neurons, the number cells containing PGP 9.5 (treated as a pan-neuronal marker) in the MG and SG in each animal was evaluated. Cells were counted in 50 ganglia (of each type) located on at least 10 slides (sections of colon were located at least 200 µm apart).

For statistical analysis, the Anova test (Statistica 13, StatSoft, Inc., Cracow, Poland) was used, and the differences were considered statistically significant at *p* < 0.05.

## 3. Results

During the present study, neurons positive for every studied neurotransmitter (VIP, GAL, VAChT and SP) were found in the submucous and myenteric ganglia of the mouse glandular stomach in all animal groups. The amount of labeled neuronal cells was different depending on the active substance and type of ganglion studied (Table 2, Figure 1 and Figure 2).

In the control group, the most numerous neuronal populations in both the MG and SG were neurons immunoreactive to VAChT. The percentage of them in the MG amounted to 54.41 ± 0.41% of all PGP-9.5-positive cells, and in the SG, this value achieved 51.70 ± 0.64% (Table 2). GAL- and/or VIP-positive neurons were less numerous. The presence of these neuronal factors was found in about one-third of all neurons labeled with PGP 9.5 (Table 2, Figure 1 and Figure 2). The least numerous were SP-immunoreactive neurons. Their percentage amounted to 20.73 ± 1.43% of all PGP-9.5-positive neurons in the MG and 14.98 ± 0.84% in the SG.

Generally, under the influence of bisphenols, the percentage of neurons immunoreactive to GAL, VIP and/or SP increased; however, the level of the observed changes depended on the type of enteric ganglia, the type of administrated bisphenol and its dose level (Figure 1 and Figure 2).

In the case of BPA, all changes noted after the administration of lower and higher doses (BPAI and BPAII groups) were statistically significant (Table 2, Figure 1). In the BPAI group, the percentage of neuronal cells immunoreactive to GAL increased from 32.84 ± 0.456% to 37.89 ± 0.612% and from 30.37 ± 0.9% to 35.32 ± 0.652% of all PGP-9.5-positive neurons in the MG (Figure 1—first row) and SG, respectively. Similar changes were observed in the case of the VIP-positive perikarya, the percentage of which increased to 41.4 ± 0.403% in the MG (Figure 1—second row) and to 35.08 ± 1.151% in the SG. In the BPAI group, an increase in the population size of the SP-immunoreactive neurons was also noted. Their percentage amounted to 31.44 ± 3.47% in the MG and 27.92 ± 0.7% in the SG (Figure 1—third row).

In the BPAII group, changes were more visible. The population size of the GAL+ neurons increased to 43.6 ± 0.816% and 38.08 ± 0.835% in the MG (Figure 1—first row) and SG, respectively. In turn, the percentage of neurons immunoreactive to VIP amounted to 44.24 ± 1.022% of all PGP-9.5-positive cells in the MG (Figure 1—second row) and 42.82 ± 0.184% in the SG. Slightly smaller changes were observed in the population of neurons containing SP, the percentage of which amounted to 36.7 ± 2.08% in the MG and 33.35 ± 2.16% in the SG (Figure 1—third row, Table 2).

In the case of BPS, changes in the neurochemical characteristics of the enteric neurons were slightly less visible than those noted under the impact of BPA (Table 2, Figure 2). In the BPSI group, statistically significant changes were noted only in the percentage of VIP-positive neurons in the MG (increasing from 35.85 ± 0.84% to 39.76 ± 0.571%) (Figure 2—first row) and cells containing SP in the SG (from 14.98 ± 0.84 to 23.52 ± 1.18%) (Figure 2—third row, Table 2).

More visible changes were noted in the BPSII group (Table 2, Figure 2). Statistically significant increases in both types of the enteric ganglia were noted in the case of GAL-positive neurons (to 35.27 ± 0.57% in the MG (Figure 2—second row) and to 35.01 ± 1.00% in the SG) and neurons containing SP (to 28.03 ± 1.3% in the MG and to 30.34 ± 2.56% in the SG (Figure 2—third row)). In turn, higher dose of BPS caused the statistically significant increase in the percentage of VIP-immunoreactive neurons only in the MG (from 35.85 ± 0.84% to 39.44 ± 0.938%) (Figure 2—first row).

Contrary to other neuronal populations studied in the present investigation, the percentage of neurons containing VAChT decreased under the influence of both bisphenols used (Table 2) After the administration of a lower dose of BPA, VAChT was noted in 48.05 ± 0.72% of PGP-9.5-positive cells in the MP and 46.52 ± 0.98% in the SG. Changes were more visible under the impact of a higher dose of BPA, where the percentage of VAChT-positive cells achieved in 39.69 ± 0.38% and 36.62 ± 1.43% of PGP-9.5-like immunoreactive cells in the MG and SG, respectively. The percentage of VAChT-positive neurons noted after the administration of a lower dose of BPS amounted to 49.88 ± 0.97% in the MG and 48.59 ± 0.24% in the SG, and they are generally similar to those noted under the impact of a lower dose of BPA. In turn, in animals treated with a higher dose of BPS, the changes were less visible than those noted under the impact of a lower dose of BPA because the percentage of neurons containing VAChT amounted to 45.32 ± 1.04% of PGP-9.5-like immunoreactive cells in the MG and 44.18 ± 1.06% in the SG.

During the present investigation, in addition to the effect of bisphenols on the neurochemical profiles of neurons, the impact of these compounds on the total number of enteric neurons was observed. The changes consisted in the decrease in the number of neurons, but they were visible only under the impact of higher doses of bisphenols.

In the MG (Figure 3) of the control mice, the mean number of neurons achieved 837.7 ± 27.01. Lower doses of bisphenols did not change this number statistically significantly because, in the BPAI group, values amounted to 796.6 ± 16.38 cells and, in the BPSI group, 810.9 ± 38.68. A statistically significant decrease in the total number of neurons in the MG was noted in the animals exposed to a higher dose of BPA and BPS, where these values achieved 687.0 ± 29.82 and 723.6 ± 15.53, respectively.

In the SG (Figure 4) of the control animals, the mean total number of the enteric cells amounted 267.7 ± 10.56. In the animals treated with a lower dose of BPA, this value achieved 256.9 ± 5.60, and in animals treated with a lower dose of BPS, 259.9 ± 7.16. These both values were not statistically significantly different from the number of neurons observed in the control animals. Higher doses of both bisphenols caused a statistically significant decrease in the number of neurons located in the SG. The number of neurons achieved 229.1 ± 6.94 and 233.3 ± 7.18 in mice exposed to BPA and BPS, respectively.

## 4. Discussion

The obtained results showed a high number of GAL, VIP, SP- and/or VAChT-positive neurons in the ENS of the mouse stomach, which is in agreement with previous studies describing the importance of these substances in this part of the nervous system in several mammalian species, including humans [16,23,33,34,35].

It is also known that the neuronal factors included in this study play multidirectional roles in the regulation of many aspects of the gastrointestinal tract physiology, including, among others, motility, secretory activity, digestion and absorption processes, blood flow and homeostasis maintenance [11,23,33,36,37]. Moreover, due to the fact that these neuronal factors synthetized in the enteric neurons may affect the immune cells, and are involved in the neuroprotective and/or adaptive reactions [33,38,39], they also take part in the answer of the organisms to pathological and toxic factors, especially those entering the body through food.

One of such substance are bisphenols, which may affect various internal organs and systems [1,8]. It is known that the degree of exposure to bisphenols clearly depends on the part of the world [1,2], and their toxicity depends on many factors, for example, on the probiotics used [40].

The knowledge of the impact of bisphenols on the ENS is rather scant and limited, mainly limited to the influence of BPA on the digestive tract of the domestic pig [10,11,16]. Comparing the results obtained during the present study with the observations concerning the influence of BPA on the gastric ENS in the domestic pig, some differences are well visible [16]. These differences concern both the number of neurons containing particular factors under physiological conditions and the severity of changes noted under the impact of BPA. The discrepancies may result, first of all, from significant disparities in the anatomy of the stomach between the mouse and domestic pig, as well as the differences in the organization of the ENS. It should be underlined that the interspecies differences in the organization of the ENS and the neurochemical characterization of the enteric neurons are commonly known [11,17,23].

Moreover, the differences between the results obtained in studies on the porcine stomach [16] and the present investigation may result from the fact that drastically different doses were studied in both of these cases. Previous studies used lower doses of BPA which were close to doses considered by the European Food Safety Authority (EFSA) as a tolerable daily intake (TDI) dose of BPA in humans [16]. It is relatively well-known that the levels of doses considered harmful varies between mammalian species, resulting from the various metabolisms of BPAs in particular groups of animals [41,42,43]. The aim of the present study was to compare the influence of BPA and BPS in the mice; therefore, the doses used were tailored to the rodents. In the light of previous studies, the dose of BPA at the level of 5 mg/kg b.w. is regarded as the no-observed-adverse-effect-level (NOAEL) dose, and the dose at the level of 50 mg/kg b.w. is regarded as lowest-observed-adverse-effect-level (LOAEL) dose in mice [29,30,31]. It should be pointed out that these doses are much higher than the doses considered safe for humans, which the European Union legislation determined at the level of 4 μg/kg b.w./day [44].

However, some similarities between the observations in the domestic porcine stomach [16] and the present study are visible. In both investigations, BPA caused the increase in the percentage of neurons containing VIP, GAL and/or SP and a decrease in the number of VAChT-positive cells, which may suggest the similar toxic mechanism of BPA in the enteric neurons of mice and pigs. Bisphenols caused the increase in the number of the gastric neurons immunoreactive to SP, with a simultaneous decrease in the number of VAChT-positive cells, which has been noted both in the present study and previous investigations [16]. It is interesting because SP often occurs in cholinergic neurons, and both substances act in a similar way (stimulating) on the muscles of the digestive tract [23]. Therefore, the mechanisms of such situation are unknown, but are probably connected with changes to the neurochemical profiles of the neurons, which increase the production of the SP along with the simultaneous inhibition of acetylcholine synthesis. This fact strongly suggests that the SP may play important roles under the impact of toxic factors, which may result from its neuroprotective properties and its influence on the immune cells [45,46,47]. In turn, acetylcholine, which is the main factor regulating the intestinal motility in physiological conditions [23], seems to be less important under the impact of toxic factors.

The current knowledge about the effect of BPS on the enteric nervous system is even more limited. It is known that BPS may affect the ENS in the mouse colon [32] and change the number of nitrergic enteric neurons in the stomach [17]. Comparing the results of previous studies to this investigation, differences in the effects of bisphenols on the ENS between the stomach and the colon can be found [32]. The most pronounced of them is the effect of bisphenols on VAChT–positive neurons. In contrast to the present study, where bisphenols reduced the number of VAChT-immunoreactive enteric neurons in the stomach, an increase in the number of such neurons in the mouse colon has been found under the influence of BPA and BPS [32].

The reason for such a situation is not clear. It may probably result from the differences in the exact roles of the neuronal factors, depending on the intestinal segment, and differences between the character of the whole of the processes occurring in the stomach and the colon. It should be pointed out that bisphenols affect the particular segments of the digestive tract to varying degrees, which results from their metabolisms [41,42,43,48]. It is known that the colonic wall is affected, not only by free bisphenols, but also by their metabolites formed in the small intestine, which also have toxic effects [48,49]. Moreover, the absorption of bisphenols takes place in the colon [48]. Perhaps this fact is the cause of the different reaction of the enteric neurons.

In turn, the results obtained in the previous studies, in which the influence of BPA and BPS on nitrergic enteric neurons in the stomach has been noted [17], have been confirmed by this study. Namely, the present results show that the changes noted in the number of VIP neurons are generally similar to those noted in the case of nitrergic neurons, which results from the fact that both substances have similar roles in the regulation of the intestinal motility and usually colocalize in the same neurons [23].

The results obtained in the present study have shown that both BPA and BPS may affect the neurochemical coding of neurons in the ENS of the mouse stomach, but changes are more visible under the impact of BPA. Therefore, the present investigation confirms that both bisphenols have similar activity. On the other hand, the present study does not show the stronger toxic effects of BPS, known from some previous observations [6].

Interestingly, the lower dose used in this study in the mice is regarded as the NOAEL dose [29,30], i.e., a dose at which no symptoms of poisoning are visible. However, even such a dose induces some changes in the expression of nervous active substances in the ENS of the mouse stomach, which suggests that it is not neutral for the living organism.

The exact justification for the observed changes may involve several issues. First of all, they may be connected with the direct neurotoxic effects of bisphenols on the enteric neurons. This thesis is supported by the fact that the neurotoxic effects of bisphenols, consisting in the disturbances in synaptogenesis and the development of neuronal cells, are relatively well-known in other parts of the nervous system [50,51,52]. Moreover, the decrease in the total number of enteric neurons noted under the impact of higher doses of both bisphenols observed in this study seems to confirm the neurotoxic effect of bisphenols in the ENS.

Considering that, the increase in the percentage of nerve cells immunoreactive for GAL, VIP and/or SP in the ENS of the mouse stomach may be connected with the neuroprotective and/or adaptive processes in which the studied neuronal substances are involved. This argument is supported by the fact that GAL, VIP and SP are involved in neuroprotective reactions that occur in response to various pathological factors, such as inflammation, neoplasm or intoxication [16,20,23,33,34,35].

Moreover, BPA changes the balance of calcium ions, which can strongly affect the functioning of the nervous tissue [53]. Calcium ions take part in the activation and secretion of hormones and neurotransmitters, as well as being involved in the conduction of nerve impulses [54]. In addition, calcium maintains homeostasis due to the fact that it ensures the proper permeability of cell membranes, determines the proper blood clotting process and reduces inflammation processes in the organism [50,54]. Therefore, changes in the expression of neuronal factors in the ENS can be the first sign of bisphenol-induced disturbances in calcium ion balance and/or subclinical inflammatory processes [55,56,57].

In turn, the lack of statistically significant changes in the total number of neurons under the influence of lower doses of bisphenols strongly suggests that not all observed changes result from the neurotoxic properties of bisphenols and are connected with the death of neuronal cells.

Therefore, the changes observed during the present research may be an effect of disturbances in the motility of the gastrointestinal tract after the exposure to bisphenols. Previous studies report that BPA causes a relaxing effect on the intestinal smooth muscles, and all neuronal factors included in the present investigation are involved in the regulation of the intestinal motility [12,23,35]. Interestingly, bisphenols caused the increase the number of neurons containing both the strong relaxant factor (VIP) and factors taking part in the contraction of the intestinal muscles (SP). The reason for such a situation is not clear, but similar observations has been made in the digestive tract of the domestic pig and mouse colon under the impact of bisphenols [10,11], as well as in the porcine ENS under the impact of other pathological factors [58]. It may be connected with other properties of the VIP and SP, which also take part in the modulation of the immune cell functions [46,59].

## 5. Conclusions

To sum up, the present study, for the first time, compares the influence of BPA and BPS on the enteric neurons in the stomach. The changes in the percentage of neurons containing VAChT, VIP, GAL and/or SP noted in the present study under the impact of BPA and BPS indicate that both bisphenols may affect the enteric nervous system in the mouse stomach. The main contribution and most important conclusion of the present study is that BPS affects the ENS in the mouse stomach in a similar way as BPA, causing the increase the percentage of neurons containing VIP, GAL and/or SP and the decrease in the number of cholinergic neurons. Although the severity of the observed changes noted under the impact of BPS is less visible than in those noted under the impact of BPA, the obtained results show that both substances are not neutral for the enteric neurons.

Further conclusions resulting from this work are as follows: (1) both bisphenols change the neurochemical characterization of the gastric enteric neurons, even in relatively low doses, namely the no-observed-adverse-effect-level (NOAEL) dose that, in the previous studies, did not produce adverse effects; (2) a clear neurotoxic effect manifested by a decrease in the total number of neurons was also noted in animals treated with the higher doses of bisphenols, and was similar in the case of BPA and BPS; (3) changes in chemical characterization of the enteric neurons observed under the influence of lower doses of bisphenols were not accompanied by a decrease in the total number of neurons, which suggests that these changes are not related to the neurotoxicity of bisphenols, but most likely result from adaptive reactions within the gastrointestinal tract; (4) the changes in the percentage of the enteric neurons containing various neuronal factors suggest that bisphenols affect various classes of the enteric neurons performing different functions, which confirms the multidirectional influence of both bisphenols studied on the living organisms.

Of course, many aspects connected with the influence of BPA and BPS on the enteric neurons in the gastric wall still remain unclear, and the explanation of the exact mechanisms of these influences requires further research.

## Figures and Tables

**Figure 1 ijerph-20-05125-f001:**
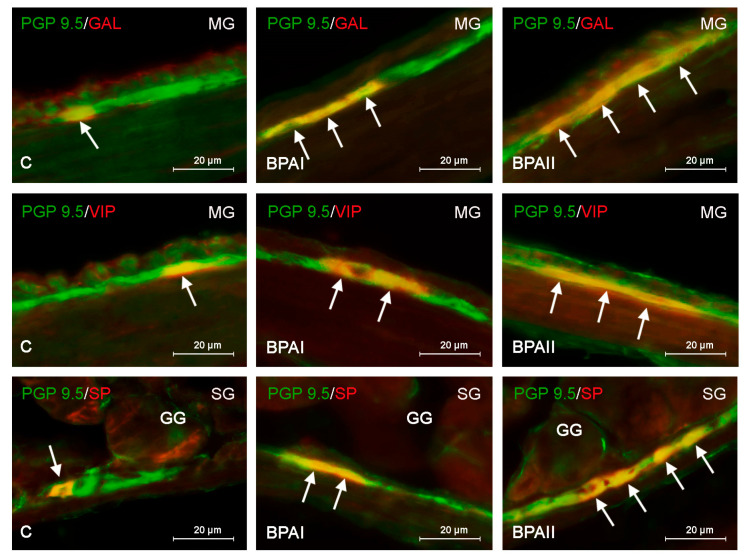
Neurons labeled with protein gene product 9.5 (PGP 9.5)—used as pan-neuronal marker—and other substance studied: galanin (GAL), vasoactive intestinal polypeptide (VIP) or substance P (SP) in the myenteric plexus (MG) or submucous plexus (SG) of the mouse stomach in the control group (C) and the groups of animals receiving a low dose of bisphenol A (BPAI) and a high dose (BPAII). In the third row, the gastric glands (GGs) from the mucosal layer are visible. The pictures were created by overlapping both stainings. The arrows point to cells that were immunoreactive for the studied substances.

**Figure 2 ijerph-20-05125-f002:**
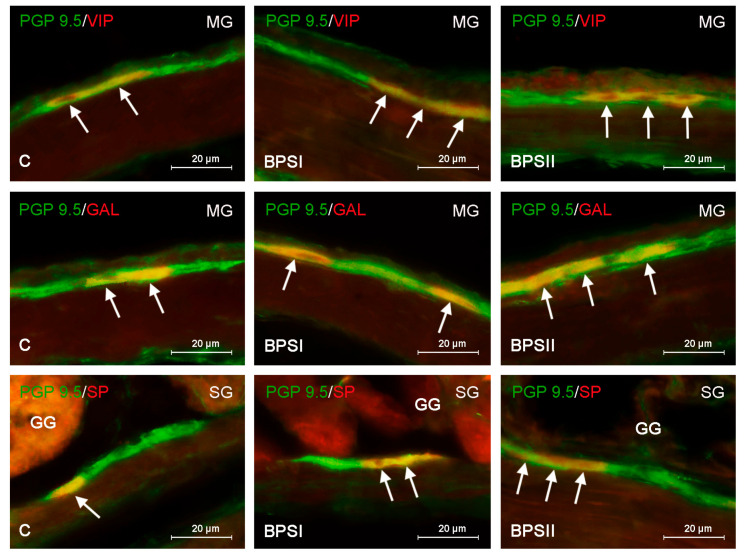
Neurons labeled with protein gene product 9.5 (PGP 9.5)—used as pan-neuronal marker—and other substance studied: galanin (GAL), vasoactive intestinal polypeptide (VIP) or substance P (SP) in the myenteric plexus (MG) or submucous plexus (SG) of the mouse stomach in the control group (C) and the groups of animals receiving a low dose of bisphenol S (BPSI) and a high dose (BPSII). In the third row, the gastric glands (GGs) from the mucosal layer are visible. The pictures were created by overlapping both stainings. The arrows point to cells that were immunoreactive for the studied substances.

**Figure 3 ijerph-20-05125-f003:**
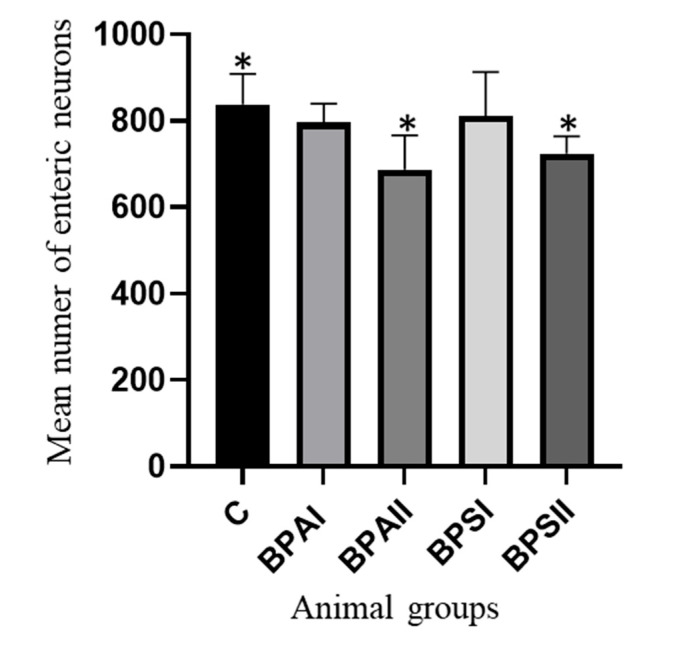
Mean number of the enteric neurons (±SEM) counted in 50 gastric myenteric ganglia (MG) in control animals (C), animals treated with BPA in low (BPAI) and high doses (BPAII) and animals exposed to BPS in low (BPSI) and high (BPSII) doses. Statistically significant differences between particular groups of animals and the control group are marked with *.

**Figure 4 ijerph-20-05125-f004:**
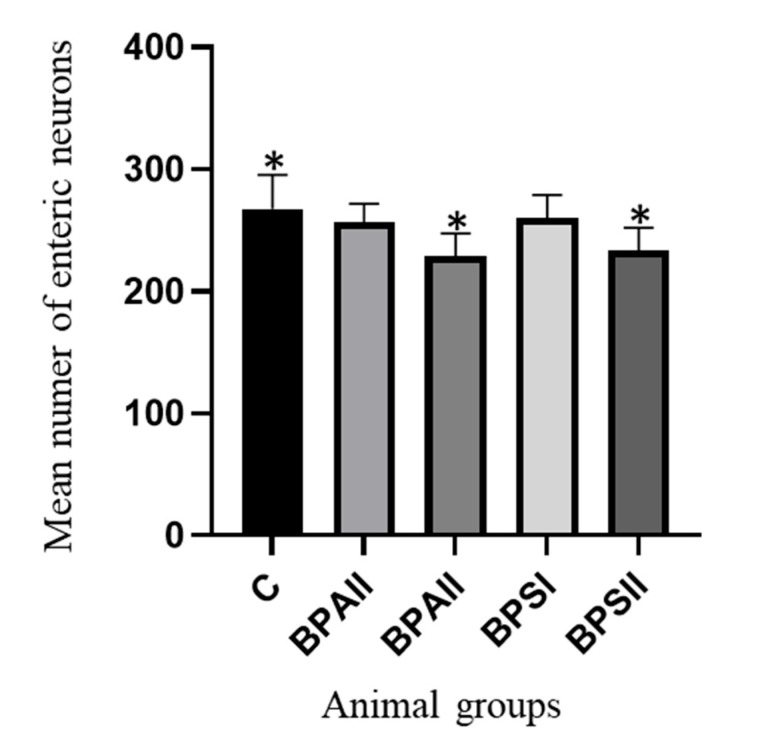
Mean number of the enteric neurons (±SEM) counted in 50 gastric submucous ganglia in control animals (C), animals treated with BPA in low (BPAI) and high doses (BPAII) and animals exposed to BPS in low (BPSI) and high (BPSII) doses. Statistically significant differences between particular groups of animals and the control group are marked with *.

**Table 1 ijerph-20-05125-t001:** List of antisera and reagents used in immunohistochemical investigations.

Primary Antibodies					Use of Antibodies on Porcine Tissues in Previous Studies
Antigen	Code	Host Species	Working Dilution	Supplier	
PGP 9.5	7863-2004	Mouse	1:1000	Biogenesis Ltd., Poole, UK	[17,32]
VIP	VA 1285	Rabbit	1:2000	Enzo Life Sciences; Farmingdale, NY, USA	[32]
GAL	T-5036	Guinea Pig	1:2000	Peninsula Labs., San Carlos, CA, USA;	[32]
SP	8450-0505	Rat	1:1000	Bio-Rad (AbD Serotec), Kidlington, UK	[32]
VAChT	H-V006	Rabbit	1:2000	Phoenix Pharmaceuticals, Inc., Burlingame, CA, USA	[32]
Secondary Antibodies					
Reagents			Working dilution	Supplier	
Alexa fluor 488 donkey anti-mouse IgG			1:1000	Invitrogen, Carlsbad, CA, USA	[17,32]
Alexa fluor 546 donkey anti-rabbit IgG			1:1000	Invitrogen	[17,32]
Alexa fluor 546 donkey anti-guinea pig IgG			1:1000	Invitrogen	[32]
Alexa fluor 546 donkey anti-rat IgG			1:1000	Invitrogen	[32]

**Table 2 ijerph-20-05125-t002:** The percentage (mean ± SEM) of neurons containing the chosen neuronally active substances in relation to all neurons labeled with PGP 9.5 in control animals and after administration of bisphenols.

Substance	Type of Enteric Ganglion	Animal Groups				
		C	BPAI	BPAII	BPSI	BPSII
GAL	MG	32.84 ± 0.46	37.89 ± 0.61 *	43.60 ± 0.82 *	32.95 ± 0.87	35.27 ± 0.57 *
	SG	30.37 ± 0.90	35.32 ± 0.65 *	38.08 ± 0.84 *	31.15 ± 0.70	35.01 ± 1.00 *
VIP	MG	38.85 ± 0.84	41.40 ± 0.40 *	44.24 ± 1.02 *	39.76 ± 0.57 *	39.44 ± 0.94 *
	SG	31.31 ± 0.62	35.08 ± 1.15 *	42.82 ± 0.18 *	32.30 ± 0.50	33.62 ± 1.34
SP	MG	20.73 ± 1.43	31.44 ± 3.47 *	36.70 ± 2.08 *	23.59 ± 0.98	28.03 ± 1.30 *
	SG	14.98 ± 0.84	27.92 ± 0.70 *	33.35 ± 2.16 *	23.52 ± 1.18 *	30.34 ± 2.56 *
VAChT	MG	54.41 ± 0.41	48.05 ± 0.72 *	39.69 ± 0.38 *	49.88 ± 0.97 *	45.32 ± 1.04 *
	SG	51.70 ± 0.64	46.52 ± 0.98 *	36.63 ± 1.43 *	48.59 ± 0.24 *	44.18 ± 1.06 *

GAL—galanin, VIP—vasoactive intestinal polypeptide, SP—substance P, VAChT—vesicular acetylcholine transporter, MG—myenteric ganglia, SG—submucous ganglia, C—control group, BPAI—animals that received BPA at a dose of 5 mg/kg b.w./day, BPAII—mice exposed to BPA at a dose of 50 mg/kg b.w./day, BPSI—animals treated with BPS at a dose of 5 mg/kg b.w./day, BPSII—mice that received BPS at a dose of 50 mg/kg b.w./day. Statistically significant values (*p* ≤ 0.05) different from the values in the C group are marked with *.

## Data Availability

Data are contained within the article.

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
