# Peer review of "Changes Caused by Bisphenols in the Chemical Coding of Neurons of the Enteric Nervous System of Mouse Stomach"

_ijerph, 2023, doi:10.3390/ijerph20065125_

Round 1

Reviewer 1 Report

This manuscript deals with the effects of bisphenol A as compared to bisphenol S on chemical coding of enteric neuerons in the mouse stomach. It continues a series of similar studies by the same group in pig and mouse. The present results demonstrate rather small and not always significant changes in the percentages of galanin, substance P and VIP immunoreactive neurons in both myenteric and submucous plexus. There are several points which should be addressed.

Major:

1)      Please specify which portions of the stomach were studied (forestomach, glandular stomach, antrum)?

2)      Materials and methods: At least 500 PGP 9.5 positive neurons examined: does it mean that in each mouse 500 PGP 9.5 neurons in each plexus were tested for each marker? Please specify!

3)      „Percentages“ and „numbers“ of neurons were mostly used synonymously. However the figures presented are just percentages!

4)      In their paper on bisphenol effects on colonic neurons (Makowska et al., Nutrients 2023, 15:200) a significant drop of neuron numbers was reported. Such quantification would be desirable also in this study.

5)      Why was VAChT omitted which was included in previous studies (pig stomach, mouse colon)?

6)      The submucous plexus is sparse in most species (cf Anetsberger D et al., Cells Tissues Organs 2018, 206:183-195; Furness JB et al., Cell Tissue Res 2020, 382:433-445). In figures 1C1-3 and 2C1-3, it appears almost as well developed as the myenteric plexus. Please comment! Besides, the red background fluorescent structures in panels C1-3 are possibly gastric glands; this should be described in the legends.

7)      The Discussion largely revolves around generalities of enteric neurobiology and effects of bisphenols. These parts can be shortened. However, a true discussion of the present results is almost lacking. For example, it may be worthwile to compare the present values for VIP with those for NOS (ref#19) at least in the myenteric plexus where these two markers typically co-exist. Likewise, the present mouse data should be compared with those in pig (ref #18). What would be also of interest are discrepant effects in different organs, e.g., increased proportions of VAChT neurons in mouse colon (Makowska et al., Nutrients 2023) as compared to a decrease in the porcine stomach (ref #18). The latter result is also remarkable because there was a concomittant increase of SP neurons, and SP is typically colocalized with VAChT or ChAT in cholinergic excitatory motoneurons. Thus, there is ample possibilitry for specific discussion, e.g., of the possible reasons for these discrepancies (technical? Species differences?).

Minor:

8)      The lettering in Table 1 (C for control, AI and AII for low and high doses of bisphenol A) and in the micrographs (A1-3, C1-3) is somewhat confusing, C designating here SP in submucous plexus. There is no documentation of SP in the myenteric plexus. Please add!

Author Response

The authors thank the Reviewer for insightful and valuable comments allowing to improve the manuscript. All suggestions have been taken into account.

This manuscript deals with the effects of bisphenol A as compared to bisphenol S on chemical coding of enteric neuerons in the mouse stomach. It continues a series of similar studies by the same group in pig and mouse. The present results demonstrate rather small and not always significant changes in the percentages of galanin, substance P and VIP immunoreactive neurons in both myenteric and submucous plexus. There are several points which should be addressed.

Major:

  • Please specify which portions of the stomach were studied (forestomach, glandular stomach, antrum)?

During the present study the corpus of the stomach (glandular part) has been studied. Information has been added (line 14 and line 121)

  • Materials and methods: At least 500 PGP 9.5 positive neurons examined: does it mean that in each mouse 500 PGP 9.5 neurons in each plexus were tested for each marker? Please specify!

Yes, 500 PGP 9.5 – positive cells were counted in each animal, in each type of the enteric ganglia, for each other neuronal factor. Information has been added (lines 147-149)

  • „Percentages“ and „numbers“ of neurons were mostly used synonymously. However the figures presented are just percentages!

During the present study the percentage of neurons was evaluated. In materials and method the term “number” has been replaced by “percentage”.

4)   In their paper on bisphenol effects on colonic neurons (Makowska et al., Nutrients 2023, 15:200) a significant drop of neuron numbers was reported. Such quantification would be desirable also in this study.

      A quantification of a total number of neurons in the studied groups were performed and the results were added to the manuscript. Lines 154-158, 241-272

5)   Why was VAChT omitted which was included in previous studies (pig stomach, mouse colon)?

      The results from the percentage of VAChT positive neurons were obtained and added to the manuscript. (lines 228-240, table 2)

6)   The submucous plexus is sparse in most species (cf Anetsberger D et al., Cells Tissues Organs 2018, 206:183-195; Furness JB et al., Cell Tissue Res 2020, 382:433-445). In figures 1C1-3 and 2C1-3, it appears almost as well developed as the myenteric plexus. Please comment! Besides, the red background fluorescent structures in panels C1-3 are possibly gastric glands; this should be described in the legends.

The authors are in agreement with the Reviewer that submucous ganglia generally contain a fewer number of cells than myenteric ganglia. However some submucous ganglia were quite big. To the photographs such large good visible ganglia have been selected.

A description of “GG” – gastric glans was added to the figures.

7)   The Discussion largely revolves around generalities of enteric neurobiology and effects of bisphenols. These parts can be shortened. However, a true discussion of the present results is almost lacking. For example, it may be worthwile to compare the present values for VIP with those for NOS (ref#19) at least in the myenteric plexus where these two markers typically co-exist. Likewise, the present mouse data should be compared with those in pig (ref #18). What would be also of interest are discrepant effects in different organs, e.g., increased proportions of VAChT neurons in mouse colon (Makowska et al., Nutrients 2023) as compared to a decrease in the porcine stomach (ref #18). The latter result is also remarkable because there was a concomittant increase of SP neurons, and SP is typically colocalized with VAChT or ChAT in cholinergic excitatory motoneurons. Thus, there is ample possibilitry for specific discussion, e.g., of the possible reasons for these discrepancies (technical? Species differences?).

The discussion has been reedited according to the suggestion of the Reviewer. Previous studies on the bisphenols and the ENS in the pig and mice have been have been closely compared with the present results. Fragments about general neurochemistry of the ENS and general toxicity of bisphenols have been removed (lines 278-376)

Minor:

8)      The lettering in Table 1 (C for control, AI and AII for low and high doses of bisphenol A) and in the micrographs (A1-3, C1-3) is somewhat confusing, C designating here SP in submucous plexus. There is no documentation of SP in the myenteric plexus. Please add!

The figures and their legends have been changed. On the figures authors have changed “A1,A2,A3” etc. to the group names to make it more understandable for the reader. In the figures authors have shown only part of the results, given figures for all the results would require at least two more figures. Therefore, the authors choosed the best photographs from the myenteric ganglion and they were obtained from neurons positive for VIP and GAL. If in the Reviewer opinion more figures should be added the authors can compose 2 more figures to show all the results obtained in the present study.  

Reviewer 2 Report

This study aims to contribute to determining the influence of BPA and BPS of neuronal populations in the gastric ENS in the rodents.

Authors must clarify and describe more clearly aspects related to the contextualization and justification of the study, as well as some aspects related to the results obtained.

For a better understanding, both the name of each group and the distribution of the animals according to the study concentrations should be clarified.

Justify the use of the concentrations used based on the maximum permitted limits included in current regulations.

Due to the insolubility of BPA, a diluent had to be used to administer it in drinking water. Indicate if this was the case and if so, whether a control group was included.

The objective of the work should not be repeated in the methodology (line 85).

Slaughter by decapitation requires prior stunning. Clarify the methodology in accordance with current regulations (line 88).

There are 2 tables 1 in the manuscript (line 105 and 128).

It is recommended to redo Tables 1 (both of them) to make them more understandable and accessible.

In Figure 2 the same magnification is indicated in all the images and yet it seems that they are not all at the same magnification. For example, the image A3 and C2. Review and clarify.

The results should be developed in a clearer and more orderly manner, using the tables and figures included.

The authors must highlight with greater clarity and emphasis what they consider to be the main contributions of their study.

On what are the authors based to affirm the following sentence? “To sum up the present study clearly shows that both BPA and BPS may affect the enteric nervous system in the mice stomach” (line 251).

It is recommended to extract and define more clearly the main conclusions of the study.

Authors must include the regulations followed according to the use of animal models, as well as the corresponding authorizations to carry out the experimental study.

Author Response

The authors thank the Reviewer for insightful and valuable comments allowing to improve the manuscript. All suggestions have been taken into account.

This study aims to contribute to determining the influence of BPA and BPS of neuronal populations in the gastric ENS in the rodents.

Authors must clarify and describe more clearly aspects related to the contextualization and justification of the study, as well as some aspects related to the results obtained.

Introduction has been supplemented with additional information as suggested by the Reviewer (lines 65-85)

For a better understanding, both the name of each group and the distribution of the animals according to the study concentrations should be clarified.

Materials and methods has been supplemented with the exact description of animal groups used in the study (lines 100-105)

Groups names have been changed from “AI”, “AII”, “SI” and “SII” to “BPAI”, “BPAII”, “BPSI” and “BPSII”

Justify the use of the concentrations used based on the maximum permitted limits included in current regulations.

It is impossible to compare the maximum permitted limits for humans to the NOAEL and LOAEL doses established for mice. For humans the European Food Safety Authority (EFSA) temporarily reduced the TDI for BPA to 4 μg/kg b.w./day because a dose of 0.05 mg/kg b.w./day (previously considered as low dose of BPA) may cause changes in the immune system, although the final decision depends on further studies. It is also known that the dose used in the present study is higher than the average exposure of humans during everyday life but humans may be exposed to even higher concentrations of BPA in some situations. For example, previous studies have reported that the amount of BPA leached from dental fillings may amount to 30 mg/day. However, because the study was performed on mice, not humans, the authors decided to focus on the studied two doses – lower dose of used in the present study (5 mg/kg b.w) is regarded as no observed adverse effect level (NOAEL) dose for BPA, and higher dose (50 mg/kg b.w) – as lowest observed adverse effect level (LOAEL) dose for BPA in mice. So, the authors assumed that the higher dose of BPA will definitely show some effects, that is why they also wanted to compare these results with the effect of BPS in such high dose and the use of lower dose was aimed to investigate the effects of bisphenols on the enteric nervous system prior to symptomatic intoxication.

However information about correlation between doses used in the study and limits of BPA intake in the European Union has been added in discussion (lines 306-313)

Due to the insolubility of BPA, a diluent had to be used to administer it in drinking water. Indicate if this was the case and if so, whether a control group was included.

To dissolve the bisphenols used 20µl of ethanol 70%. The control animals were also given an addition of 20µl of ethanol 70% to the drinking water. Information about this fact has been added in the material and methods (lines 109-112)

The objective of the work should not be repeated in the methodology (line 85).

The authors have removed the objective of the work from the methodology section. (line 117)

Slaughter by decapitation requires prior stunning. Clarify the methodology in accordance with current regulations (line 88).

The method of slaughtering was accepted by the Local Ethical Committee in Olsztyn, Poland and was chosen in accordance with IV of the European Parliament and of the Council 2010/63/EU of September 22, 2010 on the protection of animals. The animals were decapitated without prior stunning. However, the animals not in the guillotine room prior to euthanasia and decapitation took place immediately after the animal was removed from the cage. In addition, the animals' heads were immersed in liquid nitrogen immediately after decapitation. It is known that injection handling or anaesthesia before decapitating can also lead to additional stress and is therefore not considered beneficial from an animal welfare point of view. Not using anesthesia was also necessary because there was a likelihood that anesthesia or stress will disturb the chemistry of the studied tissues.

There are 2 tables 1 in the manuscript (line 105 and 128).

The mistake has been corrected.

It is recommended to redo Tables 1 (both of them) to make them more understandable and accessible.

The tables have been changed.

In Figure 2 the same magnification is indicated in all the images and yet it seems that they are not all at the same magnification. For example, the image A3 and C2. Review and clarify.

The scale is the same at every picture, the neurons found in the submucous ganglia were sometimes smaller than those found in the myenteric ganglia therefore it is shown at the micrographs. If the Reviewer is not satisfied with these figures, the authors may change the micrographs on which the neurons are slightly smaller than on the rest of the photographs.

The results should be developed in a clearer and more orderly manner, using the tables and figures included.

The results chapter has been completely reedited. The authors hope that now the results are clearer.

The authors must highlight with greater clarity and emphasis what they consider to be the main contributions of their study.

Suggestions of the Reviewer have been taken into account (lines 426-431)

On what are the authors based to affirm the following sentence? “To sum up the present study clearly shows that both BPA and BPS may affect the enteric nervous system in the mice stomach” (line 251).

The changes in the neurochemical characterization of the enteric neurons noted in the study indicate that bisphenols affect the enteric nervous system. The sentence has been reedited (lines 423-426)

It is recommended to extract and define more clearly the main conclusions of the study.

Conclusions has been reedited. The authors hope that now they are clear (lines 422-447)

Authors must include the regulations followed according to the use of animal models, as well as the corresponding authorizations to carry out the experimental study.

The use animals in the study and all experimental activities received approval of the Local Ethical Committee in Olsztyn, Poland and are in agreement with the laws in force in Poland and European Union. Additional information has been added in materials and methods (lines 91-98)

Reviewer 3 Report

I would suggest that the authors comment on the regulatory aspect of BPA and BPS (banned, restricted, etc) according to the Regulations in the EU, and worldwide as well. 

 Please change the aim of the study: present research will help to determine whether BPS 65 is less toxic than BPA and whether it should be used as a substitute for BPA in the plastic 66 industry. Namely, it is confirmed that BPS is not an appropriate alternative so you can adjust the aim accordingly. 

It is a pity that the authors used only two dose levels in the experiment. use of three doses would provide insight into dose-effect assessment i.e. relationship and maybe the application of Benchmark dose estimation.

There is literature on the use of some probiotics in BPA toxicity, maybe it could be discussed in the manner of presented exammination:

Baralić et al. Multi-strain probiotic ameliorated toxic effects of phthalates and bisphenol A mixture in Wistar rats. Food and Chemical Toxicology. 2020;143:111540.

Author Response

The authors thank the Reviewer for insightful and valuable comments allowing to improve the manuscript. All suggestions have been taken into account.

I would suggest that the authors comment on the regulatory aspect of BPA and BPS (banned, restricted, etc) according to the Regulations in the EU, and worldwide as well. 

Please change the aim of the study: present research will help to determine whether BPS 65 is less toxic than BPA and whether it should be used as a substitute for BPA in the plastic 66 industry. Namely, it is confirmed that BPS is not an appropriate alternative so you can adjust the aim accordingly. 

The aim of the study was changed according to the Reviewers suggestion.

It is a pity that the authors used only two dose levels in the experiment. use of three doses would provide insight into dose-effect assessment i.e. relationship and maybe the application of Benchmark dose estimation.

The authors are in agreement with the Reviewer, that the examination of more dose levels would be interesting however, this would require scarifying more animals and in accordance with the current restrictions concerning the protection of animals the number of animals used in the study should be minimalized. Therefore the authors decided to focus on the studied two doses – lower dose of used in the present study (5 mg/kg b.w) is regarded as no observed adverse effect level (NOAEL) dose for BPA, and higher dose (50 mg/kg b.w) – as lowest observed adverse effect level (LOAEL) dose for BPA in mice. So, the authors assumed that the higher dose of BPA will definitely show some effects, that is why they also wanted to compare these results with the effect of BPS in such high dose and the use of lower dose was aimed to investigate the effects of bisphenols on the enteric nervous system prior to symptomatic intoxication.

There is literature on the use of some probiotics in BPA toxicity, maybe it could be discussed in the manner of presented exammination:

Baralić et al. Multi-strain probiotic ameliorated toxic effects of phthalates and bisphenol A mixture in Wistar rats. Food and Chemical Toxicology. 2020;143:111540.

The authors are in agreement with the Reviewer that toxicity of bisphenols depends on many factors. The influence of probiotics on this toxicity is very interesting issue. Although the main subject of the manuscript concerns the other issues, the authors discussed proposed by the Reviewer literature (line 288).

Round 2

Reviewer 1 Report

This manuscript was adequately revised.

Author Response

The author thanks the Reviewer for the positive review.

Reviewer 2 Report

Authors have responded adequately and satisfactorily to all comments and have provided the requested information.

Author Response

(The authors gave the same response as above.)
